# GABA-A Alpha 2/3 but Not Alpha 1 Receptor Subunit Ligand Inhibits Harmaline and Pimozide-Induced Tremor in Rats

**DOI:** 10.3390/biom13020197

**Published:** 2023-01-18

**Authors:** Barbara Kosmowska, Martyna Paleczna, Dominika Biała, Justyna Kadłuczka, Jadwiga Wardas, Jeffrey M. Witkin, James M. Cook, Dishary Sharmin, Monika Marcinkowska, Katarzyna Z. Kuter

**Affiliations:** 1Department of Neuropsychopharmacology, Maj Institute of Pharmacology, Polish Academy of Sciences, 12 Smetna St., 31-343 Krakow, Poland; 2Department of Chemistry and Biochemistry, University of Wisconsin-Milwaukee, Milwaukee, WI 53211, USA; 3RespireRx Pharmaceuticals Inc., Glen Rock, NJ 07452, USA; 4Department of Pharmaceutical Chemistry, Jagiellonian University, Medical College, 9 Medyczna St., 30-688 Krakow, Poland

**Keywords:** resting tremor, action tremor, GABAA alpha 1, GABAA alpha 2/3, Parkinson’s disease, essential tremor, MP-III-024, zolpidem, positive allosteric modulator, tetrabenazine

## Abstract

Treatment of tremors, such as in essential tremor (ET) and Parkinson’s disease (PD) is mostly ineffective. Exact tremor pathomechanisms are unknown and relevant animal models are missing. GABA-A receptor is a target for tremorolytic medications, but current non-selective drugs produce side effects and have safety liabilities. The aim of this study was a search for GABA-A subunit-specific tremorolytics using different tremor-generating mechanisms. Two selective positive allosteric modulators (PAMs) were tested. Zolpidem, targeting GABA-A α1, was not effective in models of harmaline-induced ET, pimozide- or tetrabenazine-induced tremulous jaw movements (TJMs), while the novel GABA-A α2/3 selective MP-III-024 significantly reduced both the harmaline-induced ET tremor and pimozide-induced TJMs. While zolpidem decreased the locomotor activity of the rats, MP-III-024 produced small increases. These results provide important new clues into tremor suppression mechanisms initiated by the enhancement of GABA-driven inhibition in pathways controlled by α2/3 but not α1 containing GABA-A receptors. Tremor suppression by MP-III-024 provides a compelling reason to consider selective PAMs targeting α2/3-containing GABA-A receptors as novel therapeutic drug targets for ET and PD-associated tremor. The possibility of the improved tolerability and safety of this mechanism over non-selective GABA potentiation provides an additional rationale to further pursue the selective α2/3 hypothesis.

## 1. Introduction

Tremors are uncontrollable, rhythmic, and oscillating movements produced by alternating or synchronous contractions of antagonistic muscles in different parts of the body. Tremor is a common symptom of many neurological diseases, such as Parkinson’s disease (PD), or can constitute an independent disease, such as essential tremor (ET). Tremor significantly hinders daily functioning and has a direct negative impact on the patient’s quality of life. Besides some progress in the treatment of those diseases, tremor remains their key, but still untreated, symptom. Currently, there are no specific and effective enough anti-tremor pharmacotherapies and we still do not understand the exact mechanism underlying tremor pathology. Therefore, there is an immediate need to focus attention on exploring new molecular pathways and searching for improved tremorolytics drug targets.

### 1.1. Essential Tremor

ET is a chronic progressive disease that increases in prevalence with age, in which postural or kinetic tremor is present with a frequency range of 4–12 Hz, affecting various parts of the body, mainly the hands, head, tongue, legs, and less frequently other parts such as the face, torso, or voice [1,2]. Pathophysiological mechanisms underlying ET are complex and poorly understood. Results of post-mortem studies are inconsistent, some of them demonstrating Lewy Body pathology [3], cerebellar Purkinje cell (PCs) degeneration [3,4,5,6] or even normal brain morphology without PCs loss [7]. Results of clinical and post-mortem studies and research in animal models indicate that brain structures such as the inferior olive nuclei, cerebellum, thalamus, and cerebral cortex forming the cerebello–thalamo–cortical network are involved, and that hyperactivity of the glutamatergic neurons of the olivocerebellar circuit seems to be crucial for evoking tremor [2,8,9,10,11]. There are several different hypotheses regarding the mechanisms of ET generation, the most popular of which are: (1) the neurodegenerative hypothesis; (2) the central oscillatory network hypothesis; and (3) the GABAergic hypothesis [2,8,11].

A lack of understanding of ET pathophysiology makes it difficult to develop effective pharmacotherapy, and there is hardly any drug specifically developed for treating tremors. Potentiating GABA-A receptors in the CNS or inhibiting beta-adrenergic systems most consistently improves tremor in the clinic. Current symptomatic treatment of ET with propranolol (beta-antagonist) and primidone (an antiepileptic drug), is ineffective in 30–50% patients and induces a large number of side effects [2,12,13,14]. Other drugs likely to be effective in ET include topiramate or benzodiazepines (clonazepam and alprazolam) and analogues of GABA (pregabalin and gabapentin). Interestingly, ethanol also binds to GABA-A receptors and decreases ET. However, the specific anti-tremor mechanisms of actions of these drugs has not been defined. In general, it seems that substances that act via GABA-A and increase the duration of receptor opening but not the drugs that increase GABA availability are associated with tremor reduction [2]. Nevertheless, the drugs used are not tremor-specific agents but have a broad range of action, are not effective enough and show multiple harmful side effects.

Another major problem in the development of anti-tremor therapies is the lack of reliable models. One of them is GABA-A α1 global or cerebellar PCs selective knock-out mice [15,16], while the most common is the harmaline (a β-carboline derivative) model of ET [2,17,18,19]. Many substances that were shown to reduce human tremors are also effective in the harmaline model (ethanol, benzodiazepines, primidone, and other GABA-A receptor potentiators) [14,17], thus validating the model’s predictive utility for human translation, though it still has drawbacks. 

### 1.2. PD Tremor

Besides akinesia, bradykinesia, and muscle rigidity, tremor at rest is one of the main symptoms of idiopathic PD, yet most resistant to currently-available therapy. Tremor is also observed in neuroleptic-induced parkinsonism [20,21]. Bradykinesia and rigidity result from the degeneration of dopaminergic nigrostriatal neurons and disturbances in the basal ganglia-thalamo-cortico-basal ganglia neuronal circuitry [22,23,24]. Neuropathological mechanisms underlying tremors are still not fully understood but seem to be distinct from other PD motor symptoms [20,21,25]. Importantly, there is no correlation between tremor magnitude and the degree of DA deficiency in the striatum and a ‘gold standard’ therapy with L-DOPA and dopamine agonists is very poor in treating tremors [26]. Other PD therapies include deep brain stimulation (DBS) of subcortical nuclei such as the ventral intermediate nucleus of the thalamus (VIM) and subthalamic nucleus (STN), which is limited to drug-resistant, advanced forms of tremor [20].

Besides DA, other neurotransmitters including GABA and glutamate in the basal ganglia have also been suggested to be involved in the mechanism of parkinsonian tremor induction [27,28,29]. Results of neuroimaging human studies have suggested that in addition to the basal ganglia and cerebellum, which are the most important structures for the generation of parkinsonian tremor, the ventral thalamus, which receives glutamatergic projections from the cerebellum and GABAergic projections from the basal ganglia, is a relay station for neurotransmission to cortical regions [30,31,32,33]. The key structures in those neuronal networks seemingly responsible for tremor express GABA-A receptors [34,35,36]. Importantly, a phase 2, open-label clinical study (NCT03000569) with 14 PD patients explored Zuranolone (SAGE-217), an investigational oral neuroactive steroid GABA-A PAM as an oral adjunct to the treatment of Parkinsonian tremor and showed a reduction of tremor symptoms after 7 days of treatment. The MDS-UPDRS Part II/III tremor score significantly improved by 40% from baseline and persisted for 7 days after drug discontinuation [37]. This report strongly supports the role of GABA-A in tremor management, although Zuranolone is not a selective GABA-A PAM as it potentiates both synaptic and extrasynaptic GABA-A receptors, enhancing phasic and tonic inhibitory currents.

Interestingly, there are reports on the putative PD tremor-reducing effect of zolpidem, a PAM of GABA-A receptors which displays a high affinity to the α1 subunit. Daniele et al. [38] described several case studies in which zolpidem administered to patients with PD as a sleep inducer showed a visible improvement in their motor symptoms (such as akinesia, rigidity, but also resting tremor). The authors hypothesized that such effects were mediated by the high density of zolpidem binding sites in the two main output structures of the basal ganglia that are abnormally overactive in PD (globus pallidus internal, GPi, and substantia nigra reticulata, SNr), through GABA-A α1 receptors.

Currently, there are no ideal models of Parkinsonian resting tremor recapitulating the full dynamics and complexity of its pathophysiological mechanisms. The only available way of studying it in rodents is the analysis of Tremulous Jaw Movements (TJMs), defined as “rapid vertical deflections of the lower jaw that resemble chewing but are not directed at any particular stimulus” [27]. TJMs are triggered by similar conditions that lead to parkinsonism in humans e.g., by striatal DA depletion (for example by tetrabenazine and reserpine) and acute or subchronic administration of typical antipsychotics (haloperidol and pimozide) [27,39], and are suppressed by known antiparkinsonian drugs, including L-DOPA and DA agonists, as well as by DBS of the STN [39,40,41,42,43]. Multiple studies have demonstrated that TJMs depend on the ventrolateral striatum, but the involvement of both “indirect” striatopallidal and “direct” striatonigral GABAergic downstream pathways has also been suggested [35,39,44,45,46,47]. 

Although ET and PD tremors have different characteristics, it is possible that they also share some similar mechanisms. There is current evidence that ET may precede the onset of PD in a subset of patients. The risk of ET is significantly increased in relatives of PD patients, suggesting a shared hereditary predisposition. As in PD, the dopaminergic deficit and Lewy bodies as well as smell deficits are observed in some ET patients [48,49,50]. Therefore, a direct comparison of ET and PD tremors, as in the present study, could provide new insight into their mechanistic commonalities.

### 1.3. GABA-A α2/3 Subunit Specific Receptors as Novel Drug Targets against Tremor

As summarized above, convergent input from anatomical, pharmacological, and neurochemical studies have strongly implicated the enhancement of GABAergic neurotransmission through GABA-A receptors as an important anti-tremor mechanism both in ET and PD [35,38,46,51,52,53,54,55]. Although enhanced inhibition by GABA-A receptor PAMs that non-selectively interact with α1, α2, α3, and α5 subunits produces anti-tremorogenic effects, such PAMs are also associated with sedation, ataxia, tolerance development, memory impairment, and abuse liability [56]. Amrutkar et al. [51] showed that, in addition to diazepam, NS16085—a GABA-A α2/3 subtype selective modulator with reduced modulation at α1 and negligible efficacy at α5 receptors [57,58]—could also suppress tremor in a model using harmaline as a tremor inducer. This finding is potentially a highly important clue in defining a new drug target for the discovery of improved medications for treating tremors. There are several selective compounds in development that selectively target α2/3-containing GABA-A receptors [58,59]. Both preclinical and, in some cases, clinical data support the proposition of improvement in the side-effect profile of these compounds relative to non-selective GABA-A receptor PAMs [56,59]. One such α2/3-selective compound, KRM-II-81, an imidazodiazepine, already showed antiseizure, antinociceptive, and anxiolytic activity in rodent models without sedation, tolerance, or abuse liability. 

Therefore, the aim of the present study is to further test the hypothesis that the potentiation of α2/3-selective GABA-A receptors will have a tremorolytic effect. In the present work, we used the α2/3-selective PAM, MP-III-024, a compound without signficicant effect on α1- and α5-containing GABA-A receptors [59,60]. As for comparison and in relation to the described-above PD case studies, zolpidem, which selectively potentiates α1-containing GABA-A receptors [56], was used. To ensure that there is broad generality to the anti-tremor effects of this mechanism, and due to a lack of ideal models, it is necessary to employ multiple tremor inducers. The previous single study targeting GABA-A α2/3-selective mechanisms used only harmaline as an inducer. In the present study, we employed three different animal models of tremor. ET was modelled by harmaline injection while dopamine-related TJMs were induced by pimozide or tetrabenazine.

The results of this study show that zolpidem in subhypnotic doses was not effective in reducing either ET or TJM, but MP-III-024 targeting GABA-A α2/3 receptor subunits has the therapeutic potential both in the harmaline and TJMs animal models. Moreover, MP-III-024 was not sedating. We conclude that α2/3-containing GABA-A receptors comprise a new molecular target for the discovery of an improved treatment for the suppression of tremors in PD and ET patients. 

## 2. Materials and Methods

### 2.1. Animals

Adult male Wistar Han rats (Charles River, Germany), weighing 250–340 g prior to the experiment, were kept in an animal room at 12 h dark/light cycle (the light on from 7 AM to 7 PM) with a temperature of 22 ± 2 °C and relative humidity of 55 ± 5%, with 4–5 rats per cage with free access to food and water. All experiments were carried out during the light period and behavioral tests were performed during the light phase between 8 AM and 4 PM. The experiments were carried out in compliance with the Act on Experiments on Animals of 21 January 2005 amended on 15 January 2015 (published in Journal of Laws no. 23/2015 item 266, Poland), and according to the Directive of the European Parliament and of the Council of Europe 2010/63/EU of 23 September 2010 on the protection of animals used for scientific purposes. They also received the approval of the Local Ethics Committee at the Maj Institute of Pharmacology, Polish Academy of Sciences, Cracow (permission no: 271/2018, 89/2020, 90/2021). All efforts were designed to minimize both the suffering and the number of animals used. 

### 2.2. Compounds

Harmaline hydrochloride dihydrate (Sigma-Aldrich, Merck & Co. Inc., Rahway, NJ, USA) was dissolved in sterile redistilled water and administered acutely at a dose of 15 mg/kg ip. Pimozide (Sigma-Aldrich, Merck & Co. Inc., Rahway, NJ, USA), antagonist of dopamine receptors, mainly D_2_, but also D_3_ i D_4_, was dissolved in warm 0.3% tartaric acid and administered subchronically (7 injections, once a day) at a dose of 1 mg/kg. Tetrabenazine (TBZ), a reversible type 2 vesicular monoamine transporter (VMAT) inhibitor, was dissolved in 20% DMSO in 0.9% NaCl (1N HCl was added to get the drug completely into solution, according to [61,62]; final pH = 3.5) and given acutely at a dose of 2 mg/kg. This treatment procedure was based on previous experiments that employed the pimozide-induced TJMs model [47,63], see also Appendix A. Zolpidem hemitartare (synthesized and provided by Dr. M. Marcinkowska, Jagiellonian University, Medical College, Kraków, Poland), according to the previously reported procedure [64], a positive allosteric modulator of GABA-A receptors containing α1 subunit was dissolved in sterile redistilled water and administered acutely at doses of 0.34, 0.67, 1.01 mg/kg. MP-III-024 (methyl 8-ethynyl-6-(pyridin-2-yl)-4H-benzo[f]imidazo[1,5-a][1,4]diazepine-3-carboxylate, synthesized in the laboratory of J.M. Cook, University of Wisconsin-Milwaukee, (Milwaukee, WI, USA), a positive allosteric modulator selective for α2/3 subunits of GABA-A receptor, was suspended in 0.5% methyl cellulose in 0.9% NaCl and given at doses of 3.2 and 10 mg/kg in a volume of 2 mL/kg. All compounds were injected in a volume of 2 mL/kg ip. Control animals received appropriate vehicles: water, 0.3% tartaric acid, 20% DMSO in 0.9% NaCl (pH = 3.5) or 0.5% methyl cellulose in 0.9% NaCl instead of harmaline/zolpidem, pimozide, TBZ and MP-III-024, respectively. The times of administration of the above compounds are shown in Figure 1. The doses and pre-treatment times were based on the previous literature [60,65,66,67,68] and pharmacokinetic data provided by substance authors.

MP-III-024 was administered 30 min prior to harmaline, 3.5 h after the last pimozide injection. Zolpidem was administered subsequently with harmaline, 3 h 45 min after the last pimozide injection or 1 h 45 min after TBZ.

### 2.3. Experimental Procedures

#### 2.3.1. Harmaline-Induced Tremor and Locomotor Activity Analysis

The measurement of tremor and locomotor activity was performed automatically using Force Plate Actimeters (FPA; BASi, West Lafayette, IN, USA) according to [54,69,70], and started immediately after harmaline administration. FPA consists of a measuring cage placed in a sound-attenuating chamber. Four force transducers placed under the corners of the measuring cage‘s floor allow for recording the animal position on a Cartesian plane, tracking its movement across the floor and measuring the force exerted on the plate. Data were collected during time units of 10.24 s (“frames”) with the sampling frequency of 100 points/s. Tremor was analyzed using Fast Fourier Transform on each frame of the experiment. The resulting power spectra were subjected to logarithmic transformation (log10) and averaged over one (experiments with zolpidem) or two (study with MP-III-024) consecutive 180-frame series creating two intervals (0–30.72 and 30.73–61.44 min; to simplify, further times are designated as 0–30 and 30–60 min) to give the following parameters: AP1—averaged power in frequency band I (0–8 Hz), AP2—averaged power in frequency band II (9–15 Hz), Tremor Index (TI)–the difference in power between AP2 and AP1. The total distance travelled during one or two intervals, measured in millimeters, was used as a measure of locomotor activity.

#### 2.3.2. Tremulous Jaw Movements (TJMs) Analysis

The measurement of TJMs started 4 h after the last dose of pimozide or 2 h after TBZ administration. All rats used in these experiments responded to TBZ and subchronic pimozide with TJMs. Observations of rats were made in a clear Plexiglass cylinder (30 cm diameter, 40 cm high) placed in front of an angled mirror. TJMs were defined as rapid vertical deflections of the lower jaw that resembled chewing but were not directed at any particular stimulus [27]. Each individual deflection of the jaw was recorded for 10 min using a mechanical hand counter by a trained observer, who was blind to the experimental condition of the rat being observed. The observation was also video recorded for later verification with live counting.

### 2.4. Statistical Analysis

Results are presented as means ± standard errors of mean (SEM). The statistical analysis of the results was performed using STATISTICA v. 13 software (StatSoft Inc., Tulsa, OK, USA); *p* ≤ 0.05 was considered statistically significant and 0.1 ≥ *p* ≥ 0.05 were considered trends. For the results obtained using FPA (harmaline-induced tremor and locomotor activity), factorial ANOVA (zolpidem experiments) or ANOVA for repeated measures (experiments with MP-III-024; two time points: 0–30 min and 30–60 min) was used. TJMs were analyzed by factorial ANOVA. For individual comparisons between groups, the LSD post hoc test was used. The number of animals in each group is given in the figure descriptions.

## 3. Results

### 3.1. MP-III-024 Inhibits Harmaline-Induced Tremor and Increases the Locomotor Activity

In order to establish whether the GABA-A α2/3 subunit specific PAM MP-III-024 has the potential to decrease tremors related to ET, we tested its effect on harmaline-induced tremor and locomotor activity. Harmaline (15 mg/kg) caused a generalized tremor in rats, which was observed as a significant increase in the AP2 parameter (power in the range of 8–15 Hz) (Figure 2B) and the tremor index (Figure 2C) compared to the control group and both these changes persisted for the entire 60 min of tremor measurement. Harmaline additionally altered the locomotor activity of rats, measured by the total distance travelled, decreasing it in the first 30 min, and then increasing walking between 30–60 min after administration as compared to the control rats (Figure 2E). Harmaline itself lowered AP1 (power within 0–8 Hz frequency band) within the first 30 min of measurement (Figure 2D). This experiment was performed up to 60 min after harmaline injection to observe the length of effect (Figure 1A).

MP-III-024, a positive allosteric modulator of α2/3 subunit containing GABA-A receptors, given alone had no effect on AP2 and distance parameters, but showed a trend to decrease tremor index (both doses) as compared to solvent. MP-III-024 itself also increased animal mobility. The lower dose (3.2 mg/kg) extended the total distance travelled by the animals during the first 30 min of measurement and raised the AP1 parameter at the same time. The increase in AP1 was also noted for the dose of 10 mg/kg in comparison to solvent control (Figure 2D).

MP-III-024 at both doses used (3.2 and 10 mg/kg), significantly attenuated the tremor induced by harmaline, which is illustrated by the power spectrum and AP2 graphs (Figure 2B). The anti-tremor effect of MP-III-024 at the higher dose was weaker, as it affected AP2 but not the tremor index parameter and lasted for the entire 60 min of measurement. On the other hand, the effect of MP-III-024 at the lower dose was stronger and more visible, as it decreased both the AP2 and the tremor index but lasted for a shorter duration (0–30 min). Additionally, following harmaline, MP-III-024 in both doses increased the locomotor activity (distance parameter) in comparison to animals treated with harmaline alone during 0–30 min (after the dose of 3.2 mg/kg) or 0–60 min of measurement (10 mg/kg) (Figure 2E). Generally, the observed effects of MP-III-024 on harmaline tremor were not dose-dependent in the tested range.

### 3.2. MP-III-024 Inhibits the Pimozide-Induced TJMs

To determine if a GABA-A α2/3 subunit specific PAM is effective in reducing tremor in other types of tremor, pimozide was administered at a dose of 1 mg/kg once a day for 7 days. Pimozide induced TJMs in all animals. There were no differences between the effect of MP-III-024 given alone and the solvent (SOLV) control group, but MP-III-024 at both doses used (3.2 and 10 mg/kg) significantly inhibited pimozide-induced TJMs (Figure 3).

### 3.3. Zolpidem Has No Effect on Either Harmaline-Induced Tremor or TJMs after Pimozide or TBZ

To verify if GABA-A α1 subunit specific drug zolpidem could affect tremor related to ET, it was tested in a harmaline model. Harmaline (15 mg/kg), similar to our previous experiments [54,69,70], induced whole-body tremor. It appeared a few minutes after administration and was manifested by an increase in tremor power within the frequency band of 8–15 Hz (AP2) (Figure 4B), with the peak for frequencies around 10 Hz (Figure 4A), and in increasing the tremor index parameter (AP2-AP1, Figure 4C) compared to the control group for the entire 30 min of measurement. At the same time, harmaline decreased the AP1 parameter representing animal movements in the lower range of frequency (0–8 Hz) but without affecting the locomotor activity of the animals (distance) (Figure 4D,E).

Zolpidem, a positive allosteric modulator of GABA-A α1 receptors, was tested in three subhypnotic doses [65,66]. Given alone, zolpidem had no major effect on tremor or activity parameters as compared to control solvent (SOLV) group, but the profile of changes was dose-dependent. In particular, at the highest dose (1.01 mg/kg), zolpidem itself increased the tremor index, while in the middle and the highest doses (0.67 and 1.01 mg/kg) it also lowered the AP1 and distance parameters compared to controls (Figure 4C,D).

Zolpidem at the lowest (0.34 mg/kg) and the highest dose (1.01 mg/kg), administered together with harmaline, seemed to increase the power between 8–15 Hz above the harmaline-induced peak when looking at the power spectrum graph (Figure 4A), but no such effect was observed when analyzing the values of individual tremor parameters (AP2 or tremor index) (Figure 4B,C). Coadministration of zolpidem (at any dose) with harmaline had no effect on tremor parameters, AP2 and tremor index, or on locomotor activity compared to animals treated with harmaline alone.

Pimozide administered at a dose of 1 mg/kg once a day for 7 days, as well as TBZ given as a single dose of 2 mg/kg, induced TJMs in all animals subjected to those treatments. Control experiment showing Pimozide or TBZ influence on locomotor activity and catalepsy is shown in Appendix A. Zolpidem administered alone in the middle dose (0.67 mg/kg) had no effect, and coadministration with pimozide or TBZ did not affect TJMs in either model (Figure 5A,B).

## 4. Discussion

GABA-A receptors are ionotropic, rapidly-responding ligand-gated ion channels formed as heteropentamers. They can be assembled from 19 different subunits forming Cl channels, thus presenting enormous heterogeneity of receptor subtypes. The α1-containing GABA-A receptors are the most widely distributed GABA-A receptors in the brain (48% of the global gene expression for GABA-A receptors), expressed predominantly in the cerebral cortex and cerebellum, while α2 and α3 subunits are mostly present in the thalamus and molecular layers of the cerebellum [71] (see also below). Depending on the subunit composition and arrangement, as well as distribution in the various brain regions, these receptors exhibit different pharmacology and functional output of tested substances. Two GABA binding sites are formed by an α and β subunits and one benzodiazepine binding site formed by an α and γ subunits. Whereas the GABA neurotransmitter opens the Cl channel, ligands acting via the benzodiazepine binding site cannot directly open it, but only allosterically enhance (positive allosteric modulators-PAMs) or reduce (negative allosteric modulators) GABA-induced currents [56,57].

### 4.1. The Tremorolytic Effect of GABA-A α2/3 Subtype Selective Positive Allosteric Modulator MP-III-024

In this study we tested novel GABA-A α2/3 selective receptor PAM, MP-III-024 and showed that it significantly reduced the harmaline-induced tremor which is in line with the previous reports using other GABA-A α2/3 targeting but less selective substances of different pharmacological profiles [51].

So far, multiple studies showed tremorolytic effect of non-selective GABA-A α1/2/3/5 receptor PAMs both in ET patients and experimental rodent studies. Anti-tremor effects of benzodiazepines, ethanol, alprazolam, diazepam, muscimol, and used as first-line therapy for ET–primidone, strongly support a role for the GABAergic system in ET. However, their clinical use is limited due to side effects such as sedation, ataxia, tolerance development, and memory impairment [2,51,56]. With such a non-selective approach, sedation, ataxia, and dependence are related to the activity at the α1 subunit, muscle relaxation to α2, anxiolysis and analgesia and anticonvulsant effects to α2/3, and memory impairment depends on the α5 subunit [56].

Therefore, Amrutkar et al. [51] were the first to hypothesize that subtype selective GABA-A receptor modulators acting selectively via the α2 and α3 subunits may have an improved side effect profile while retaining the tremorolytic beneficial effects in ET. They showed experimentally in the animal model of ET, similar to that induced by harmaline, a tremorolytic effect of NS16085 [51], which is a GABA-A α2/3 subtype selective modulator with low level of negative modulation at α1 and negligible efficacy at α5 receptors [57,58]. By comparison of NS16085 with NS11394-GABA-A α2/3/5 receptor PAM, they showed that it was the α2/3, but not the α5, subunit which played a role in the anti-tremor efficacy [51]. Our study, using another GABA-A α2/3 selective compound, MP-III-024, supports their observations. Another study on ET patients using GABA-A α2/3/5 subtype-selective PAM, TPA023, have shown its effectiveness “superior to placebo, but not statistically significant” in reducing kinetic tremors in ET patients [72]. Therefore, GABA-A α2/3/5 subtype-selective PAMs have tremor-reducing potential both in human and in an animal ET model. The absence of α5 potentiation by MP-III-024 [63] strongly suggests that potentiation of α2/3-containing GABA-A receptors is sufficient. Since α5 has potential burdens on cognition, sedation, and tolerance [62], further study of the potential therapeutic value of α2/3-selective PAMs are warranted.

Our studies went a step further, and we tested GABA-A α2/3-specific PAM without α1 or α5 efficacy, MP-III-024, and confirmed the previous observation in harmaline-induced ET-like tremor.

That α2/3, but not α1, subunit-containing GABA-A receptors are sufficient for anti-tremor efficacy is based on the MP-III-024 pharmacological profile proven previously in vitro in functional GABA receptor assays [60] and the fact that zolpidem, which is α1-selective, was not tremorolytic and NS16085’s effectiveness was shown even with its small α1 antagonism. Additionally, other selective α2/3-receptor PAMs have shown activity in tremor patients (discussed above).

Moreover, we show for the first time that MP-III-024 was also effective not only in the ET model but also against pimozide-induced TJMs.

MP-III-024 is a positive allosteric modulator with preference for GABA-A α2 and α3 receptors relative to α1 and α5, as well as an improved metabolic profile as compared to another subtype-selective PAMs. Although MP-III-024 is an ester, it is more metabolically stable than its analog, HZ-166 [60,67]. In addition, other analogs such as KRM-II-81 that substitute a bioisostere for the ester function are known to provide markedly-improved drug-like properties [59]. MP-III-024 has a good metabolic stability, with 75% and 76% remaining after one hour of mouse or human liver microsome assay. The half-life of MP-III-024 is 141.5 min with an intrinsic clearance of 0.491 µL/min/mg and a metabolic rate of 9.815 nmol/min/mg [60]. As an ionotropic receptor ligand, its effects are observed very fast after administration. Our study timeline fits in the maximum drug bioavailability window. MP-III-024 produced significant decreases in tremor but these effects were not always dose-dependent. This finding contrasts with the dose-dependent changes in nociception observed with this compound [60]. The data show that the lowest dose of MP-III-024 tested (3.2 mg/kg) produced almost complete reversal of harmaline-induced tremor in the AP2 parameter (Figure 2B). Doses lower than 3.2 mg/kg should be tested in future studies to fully realize the dose range of activity of this compound. Nonetheless, the data suggest an important possibility that tremor is more potently controlled by α2/3-containing GABA-A receptors than nociception.

Fischer et al. [60] showed that MP-III-024 dose- and time-dependently reversed mechanical hyperalgesia (10 and 32 mg/kg) and did not affect locomotor activity or operant behavior. It was reported by Rahman et al. [68] that MP-III-024 can reduce the amount of opioids needed to control pain when dosed together with morphine, therefore showing synergistic pain reduction. As predicted, substances of this type are non-sedative, do not impair motor functions and do not develop tolerance or addiction. An example of this was described using another compound of similar pharmacology, KRM-II-81, which was disclosed as a non-sedating anxiolytic-like agent. Preclinical data demonstrated its efficacy in animal models of anxiety, depression, acute and chronic pain, epilepsy, and traumatic brain injury, along with reduced sedation, motor-impairment, tolerance development, and abuse liability [56,73]. Our results indicate that GABA-A α2/3 could become a starting point for studies to understand tremor generation and extinction mechanisms and indicate a new biological target for therapy.

### 4.2. Different Tremor Models, Same Effect—Why?

Interestingly, for comparison, we also tested MP-III-024 in pimozide-induced model of TJMs and it also generated a significant tremorolytic effect. Since MP-III-024 was effective in tremor reduction in harmaline and pimozide-induced models, GABA-A α2/3-related mechanisms could be the common point of tremor regulation. Despite that both models are induced by differently acting substances, they potentially share common effector. Thus, MP-III-024 effect is probably located downstream of striatum and at least partially dependent on GABA-A α2/3.

Tremors in ET are mostly postural or kinetic with a frequency range of 4–12 Hz, while resting tremor in PD occurs in a frequency range of 3–7 Hz. Pimozide acts as an antagonist of dopamine D_2_, D_3_, and D_4_ receptors; the 5-HT_7_ receptor and SNr and GPi appear to be a major basal ganglia output region through which the thalamo-cerebellar circuit and tremor are regulated [27]. Interaction between the basal ganglia and cerebellum was also indicated as the most important for the generation of parkinsonian tremor. This concept has been summarized as the “dimmer-switch” theory of PD tremor, which proposes that tremor is initiated in basal ganglia structures, but the cerebellum modulates its amplitude and rhythmicity [25,31,33,34]. Recent studies indicated that other brain structures, such as the ventral thalamus, which receives glutamatergic projections from the cerebellum and GABAergic projections from the basal ganglia, appear to be a region of convergence of neuronal impulses, where they are processed and relayed to the cerebral cortex. These basal ganglia-thalamo-cortical and cerebello-thalamo-cortical circuits were assumed to be completely independent, but recent data suggested some points of their contact, such as the motor cortex [30,32]. Therefore, the pathophysiology of parkinsonian tremor also needs further research.

The suggested mechanisms of harmaline-induced tremors are that it enhances synaptic activity of the climbing fibers originating in the inferior olive nuclei. Their pathological activation causes excessive glutamate release in the cerebellum [74,75] and affects GABAergic Purkinje cells. This leads to disinhibition of deep cerebellar neurons consequently increasing glutamate release in the thalamus [69]. Rhythmic activity of this structure, results in generation of action tremor, the clinical feature of ET [2,11]. The exact mechanisms underlying tremor generation by harmaline, or in ET are still unclear.

Based on the results of this study, we can only roughly hypothesize but it seems that both in pimozide- and harmaline-induced tremors and by approximation probably in ET and PD, the common tremor regulating pathways include the cerebellum, thalamus and motor cortex.

### 4.3. Lack of Tremorolytic Effect of Zolpidem in Experimental Studies

Zolpidem is the most selective known GABA-A α1 PAM used in the clinic as a hypnotic drug for the short-term treatment of sleeping problems. Previous clinical observations by Daniele et al. [76] in a pilot study with ten PD patients showed remarkable improvement of motor symptoms, surprisingly including tremor, even after a single dose of zolpidem. A case report by Farver & Khan [77] described the beneficial effects of zolpidem in patient suffering from antipsychotic-induced parkinsonian-like tremors. In addition, Růzicka et al. [78] and Hall et al. [79] reported improved patients’ UPDRS scores after zolpidem. The hypothesis behind those observations was that in PD overactivity of the GABAergic neurons in the GP can lead to overinhibition of the thalamus and the cerebral cortex. With a selective inhibition of GABA-A α1 in the main structures involved in movement disorder zolpidem might counteract it.

In contrast, a recent study by Diamond et al. [80] indicated no change in GABA-A α1 expression in the GP of PD patients as compared to healthy humans, counteracting the classical hypothesis of pathophysiology of movement disorder evoked by dopamine loss.

Besides clinical case reports and promising hypothesis and few optimistic reviews, there are limited studies attempting to validate the GABA-A α1 role in PD tremor besides the motor dysfunction. Even reports by Daniele et al. [76] and zolpidem use in the clinic were focused on motor impairment while tremor was indicated only as one of the symptoms. Recent metanalysis by Laifenfeld et al. [81] suggested repurposing zolpidem for use in slowing PD progression but their analysis was focused on dementia, falls and psychosis, not tremor. There are two registered clinical trials on zolpidem in PD (clinicaltrials.gov). One was withdrawn and second assessed motor function including tremor (NCT03621046), although the data are still not released. The same group published interesting results on zolpidem positively affecting bradykinesia and movement initiation by the measurement of beta oscillations in 17 PD patients [82] but again, not tremor.

Despite promising preliminary effects in PD patients, and an attractive working hypothesis based on GABA-A α1 expression in the key brain structures related to PD and tremor and strong selectivity of the tested drug, in our experimental study zolpidem did not affect tremor in either of the tested doses and none of the different animal tremor models. The studies of Diamond et al. [80] are in line with our suggestion that α1 subunit does not play a role in CNS-mediated PD tremor.

One of the reasons for the lack of zolpidem effectivity could be due to an overly widespread expression of GABA-A α1 in the brain to evoke specific effects on the tremor, despite its localization in key regions of interest. GABA-A receptors and associated α1 subunit are expressed in the SNr, GPi, GPe, STN, thalamus, motor cortex and many other structures, thus possibly counteracting each other function. The second, more reliable reason could be the role of advanced dopaminergic neuron degeneration and resulting adaptation/dysfunction of multiple downstream regulatory loops, as well as the involvement of other neuronal pathways affected in severe stages of late PD as it was described in the clinical case reports. The complexity and heterogeneity of patient PD tremor is not fully reflected by the available simplified rat models.

In our test models using pimozide and tetrabenazine for induction of TJM, the nigrostriatal system was only functionally and temporarily blocked. Therefore, although it is possible to evoke dopamine-related tremor in rodents even without permanent nigrostriatal lesion our results in rat models indicate that tremor induction by pimozide or tetrabenazine is not identical with human tremor and is based on mechanisms far downstream from striatum and probably not GABA-A α1 selectively–related process.

Importantly, besides just a few studies on non-human, old age primates, the available rodent models of PD-related nigrostriatal degeneration and/or proteinopathy evoking loss of dopamine (6-OHDA, MPTP/MPP+, α-synuclein, rotenone, paraquat, genetic models, etc.), even those most severe, do not exhibit tremors. Therefore, the direct role of dopamine in the pathophysiology of PD-tremors is speculative, especially given that L-DOPA treatment only partially alleviates tremor in patients. Although tremor is considered to be the most specific marker of PD, its severity has not been shown to parallel disease progression or the degree of nigrostriatal neuron loss. However, recent study showed that rest tremor correlates with reduced contralateral striatal dopamine transporter binding in PD patients [83]. Therefore, research is inconclusive in this aspect. PD is also a very heterogenous disease itself, recently being divided into multiple more specific subtypes. Thus, the tremor-dominant PD subtype has a more benign course and it is hypothesized that tremor itself could be the sign of compensatory mechanisms counteracting akinesia/bradykinesia. All these prompts to hypothesize that additional anatomical regions, downstream affected networks, and neurotransmitters contribute to parkinsonian tremor. In the exploratory small clinical study by Daniele et al. [76], the best effects of zolpidem were observed in the patients with the most advanced stages of PD. Therefore, long-term use of antiparkinsonian pharmacotherapy and system adaptation in late-phase PD patients should be taken under consideration [84].

Our results indicate that human late PD-tremor and tetrabenazine- or pimozide-induced dopamine-related rat tremor may not have overlapping pathomechanisms with respect to GABA-A α1 receptor subunit. The question remains which physioanatomical or functional aspects of the human, but not rodent, brain is responsible for tremor generation. This discrepancy could be a starting point for a search for tremor-specific mechanisms in PD. Therefore, if the tested models represent at least partially-relevant mechanisms underlying tremor generation, our results are not in line with repurposing zolpidem as an anti-tremor medication. The mechanistic role of the effects zolpidem on akinesia, bradykinesia, stiffness, and other PD symptoms should be studied otherwise.

### 4.4. Lack of Tremorolytic Effect of Zolpidem in ET Model

Our results show also that zolpidem has no influence also on harmaline-induced ET-like tremor in rats in neither of the three tested doses. In the report by Assini & Abercrombie [52], the first dose affecting motor function tested on rotarod in naive mice was 1 mg/kg of zolpidem. A dose of 0.5 mg/kg did not evoke any significant impairment. In our study, although we tested a similarly low, sub-hypnotic doses of zolpidem, it decreased locomotor activity measured in FPA as distance travelled and general rat mobility measured as AP1 parameter in the two higher doses (0.67 and 1.01 mg/kg) 30 min after treatment. The effect was dose-dependent. This is in line with known observations that GABA-A α1 subunit is responsible for sedation. Therefore, those results point out that if harmaline induced tremor is relevant to ET physiology then GABA-A α1 is not specifically involved in ET pathology. 

### 4.5. GABA-A α1 vs. α2/3 Subunit Localisation

In this study, tremor was decreased by GABA-A α2/3 but not GABA-A α1 selective PAMs. The pharmacological output of tested substances depends on receptor subunit composition and distribution in the specific brain regions. Therefore, based on our results, one could hypothesize that structures with dominant α2/3 but not α1 expression could be the regions of interest for further detailed studies of tremor.

One of the possible explanations is the dominating expression of α2 and lack of α1 GABA-A subunit in the inferior olivary complex, a structure which enhanced synaptic activity which is directly responsible for tremor generation, at least in ET and harmaline-induced models. Activation of inhibitory GABAergic transmission in this structure is probably responsible for decreasing tremor. Other potential targets lacking GABA-A α1, but with high expression of α2, are striatum and cerebellar Purkinje cell layers with no α1 or α2 subunits but numerous α3 protein signals. Strong immunoreactivity for α2- and α 3-containing GABA-A receptors has been detected in the parvalbumin positive neurons in the thalamus and molecular layers of the cerebellum [85,86,87,88,89]. Hypothetically, structures expressing α2/3 but not α1, GABA-A receptor subunits could be the target for future tremor therapies.

### 4.6. Effect on Locomotor Activity

Side effects characteristics of benzodiazepines are daytime drowsiness and impaired motor coordination. In fact, GABA-A α2/3, but not α1, specific PAMs have an improved profile in this regard. Previous studies [60] already showed that MP-III-024 did not affect locomotor activity (3.2, 10, 32 mg/kg) during 60 min of open field tests or operant behavior (3.2, 10, 32, 100 mg/kg) during 160 min in the nose-poke operant chamber test. In our study MP-III-024 alone tested in a lower dose (3.2 mg/kg) significantly increased distance travelled (walking), and all animal movements (not only locomotion but also body movements in place) quantified as AP1 parameters in both doses (3.2 and 10 mg/kg). This increased animal mobility was observed during the first 30 min of the test in FPA, which corresponds to the first 60 min after MP-III-024 injection. As compared to the results of Fischer et al. [60], their animals at the corresponding doses did not show significant increases in the two types of movement behavior analyses, although some tendencies might be visible. The FPA analysis used in our study is a much more sensitive method of quantification than open field. When MP-III-024 was administered before harmaline it also increased locomotor activity as compared to harmaline alone group. Interestingly, harmaline itself has bidirectional effects on locomotor activity in rats due to tremor generation, when analyzed directly after i.p. injection [70]. In the first 30 min of analysis, it decreases distance travelled then increases it in the following 30 min as compared to controls. It seems that harmaline treatment and induction of tremor in rats postpones their exploratory activity and it was also observed in the combined treatment group. In conclusion, MP-III-024 not only does not impair motor activity in rats but temporarily increases it in time-points mostly corresponding with exploratory activity. Further cognitive- and motivation-driven tests would bring more information on the role of GABA-A α2/3 specific mechanisms. Nevertheless, additional mechanisms facilitating movement could be of the highest interest in potential PD pharmacotherapy using selective GABA-A receptor α2/3 subtype but not α1 specific PAMs.

## 5. Conclusions

Traditional GABA-A PAM drugs have safety and tolerability concerns that include sedation, motor-impairment, respiratory depression, tolerance, and dependence. The search for improved therapeutic agents focuses on more selective ligands that potentiate GABA-A receptors [56]. There is a huge void in pharmacotherapy of tremor, therefore, it is necessary to target it as the main study subject. Here we have shown for the first time that GABA-A α2/3, but not GABA-A α1, is a promising target for tremorolytic therapies both in ET and PD tremor disorders.

Our results indicate that zolpidem, a positive allosteric modulator selective to GABA-A α1 subunits did not affect tremor induced by harmaline, tetrabenazine, or pimozide, while MP-III-024—a positive allosteric modulator selective to GABA-A receptor α2/3 subunits—decreased tremor parameters in both ET and PD related types of tremors. The effectiveness of MP-III-024 suggests at least partially overlapping mechanism of tremor pathophysiology in those two neuropathologies. The results of this work offer an important hint in elucidating tremor suppression mechanisms. The data suggest that GABA-A α2/3 should be considered a novel drug-target for PD and ET tremor therapy, with additional potential towards movement facilitation in PD.

## Figures and Tables

**Figure 1 biomolecules-13-00197-f001:**
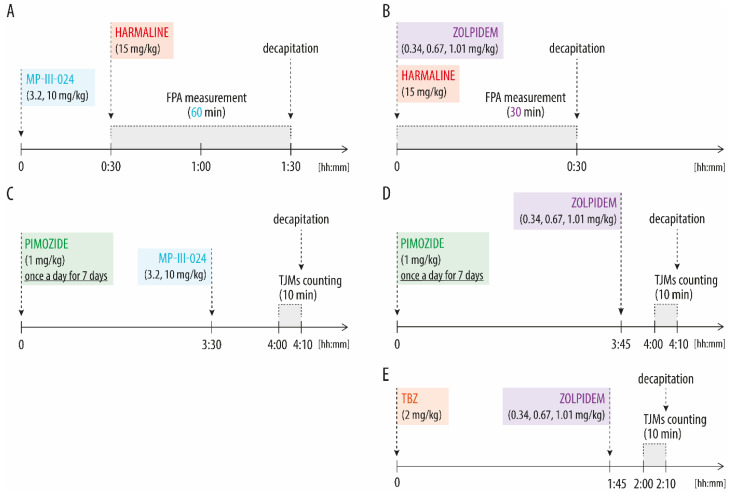
Timelines of drug administration and harmaline-induced tremor measurement (**A**,**B**) or observation and counting of TJMs induced by pimozide (**C**,**D**) or TBZ (**E**). FPA measurement of harmaline-induced tremor lasted 30 min or 60 min for experiments with zolpidem and MP-III-024, respectively. Abbreviations: FPA—Force Plate Actimeter; TBZ—tetrabenazine; TJMs—Tremulous Jaw Movements.

**Figure 2 biomolecules-13-00197-f002:**
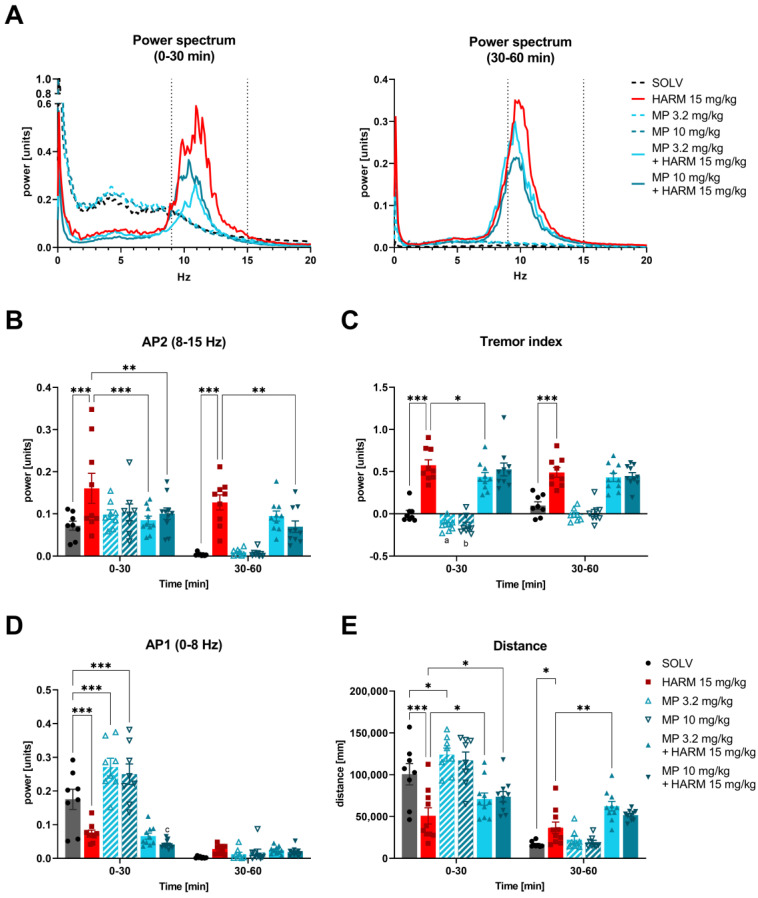
The effect of MP-III-024 (3.2, 10 mg/kg) on the harmaline-induced tremor: (**A**) power spectrum, (**B**) AP2, (**C**) tremor index, (**D**) AP1, and (**E**) locomotor activity (distance) of rats. The power spectrum within a range of 0–20 Hz, averaged over the two intervals (0–30 and 30–60 min) for all animals in each group is shown. Tremor parameter data are shown as the means ± SEM. AP1—power in the 0–8 Hz band, AP2—power in the 8–15 Hz band. The number of animals: SOLV, *n* = 8; HARM 15 mg/kg, *n* = 8; MP 3.2 mg/kg, *n* = 8; MP 10 mg/kg, *n* = 8; MP 3.2 mg/kg + HARM 15 mg/kg, *n* = 10; MP 10 mg/kg + HARM 15 mg/kg, *n* = 10. Statistics: ANOVA for repeated measures with regard to AP2 (HARM effect: F[1,47] = 27.111, *p* = 0.001; MP effect: F[2,47] = 1.459, *p* = 0.243; time effect: F[1,47] = 54.740, *p* = 0.001), Tremor index (HARM effect: F[1,47] = 217.767, *p* = 0.001; MP effect: F[2,47] = 3.576, *p* = 0.036; time effect: F[1,47] = 2.004, *p* = 0.163), AP1 (HARM effect: F[1,47] = 83.421, *p* = 0.001; MP effect: F[2,47] = 2.381, *p* = 0.104; time effect: F[1,47] = 275.344, *p* = 0.001) and Distance (HARM effect: F[1,47] = 3.537, *p* = 0.067; MP effect: F[2,47] = 5.414, *p* = 0.007; time effect: F[1,47] = 279.195, *p* = 0.001). LSD post-hoc test: * *p* ≤ 0.05, ** *p* ≤ 0.01, *** *p* ≤ 0.001, ^a^ *p* = 0.077, ^b^ *p* = 0.060 vs. SOLV; ^c^ *p* = 0.090 vs. HARM 15 mg.

**Figure 3 biomolecules-13-00197-f003:**
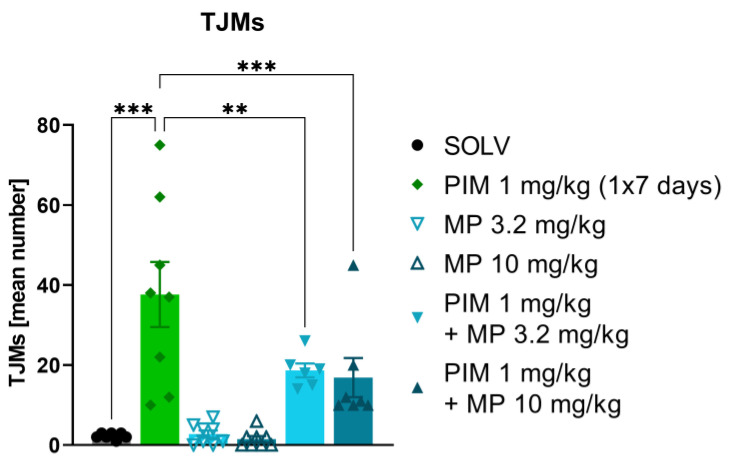
Reversal of the pimozide-induced tremulous jaw movements (TJMs) by MP-III-024 (3.2, 10 mg/kg). The data are shown as the mean (± SEM) number of tremulous jaw movements (TJMs) per 10 min observation period. The number of animals: SOLV, *n* = 7; PIM 1 mg/kg, *n* = 8; MP 3.2 mg/kg, *n* = 8; MP 10 mg/kg, *n* = 8, PIM 1 mg/kg + MP 3.2 mg/kg, *n* = 6; PIM 1 mg/kg + MP 10 mg/kg, *n* = 7. Statistics: Factorial ANOVA (PIM effect: F[1,38] = 41.736, *p* = 0.001; MP effect: F[2,38] = 3.919, *p* = 0.028) + LSD post-hoc test (** *p* ≤ 0.01, *** *p* ≤ 0.001).

**Figure 4 biomolecules-13-00197-f004:**
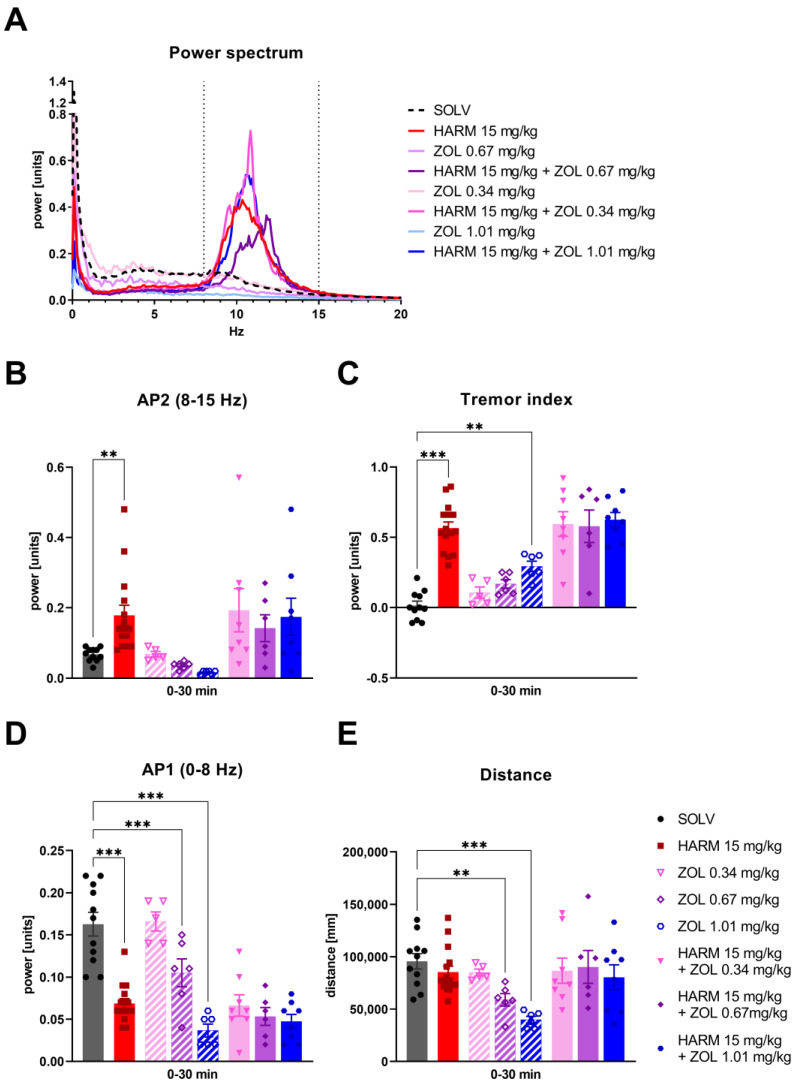
The effect of zolpidem (0.34, 0.67, 1.01 mg/kg) on harmaline-induced tremor: (**A**) power spectrum, (**B**) AP2, (**C**) tremor index, (**D**) AP1, and (**E**) locomotor activity (distance) of rats. The power spectrum within a range of 0–20 Hz, averaged over the whole measurement period (0–30 min) for all animals in each group is shown. Tremor parameters data are shown as the means ± SEM. AP1—power in the 0–8 Hz band, AP2—power in the 8–15 Hz band. The number of animals: SOLV, *n* = 11; HARM 15 mg/kg, *n* = 15; ZOL 0.34 mg/kg, *n* = 5; ZOL 0.67 mg/kg, *n* = 6; ZOL 1.01 mg/kg, *n* = 6; ZOL 0.34 mg/kg + HARM 15 mg/kg, *n* = 8; ZOL 0.67 mg/kg + HARM 15 mg/kg, *n* = 6; ZOL 1.01 mg/kg + HARM 15 mg/kg, *n* = 8. Statistics: Factorial ANOVA with regard to AP2 (HARM effect: F[1,57 = 21,391, *p* = 0.001; ZOL effect: F[3,57] = 0.546, *p* = 0.653), Tremor index (HARM effect: F[1,57] = 103.001, *p* = 0.001; ZOL effect: F[3,57] = 3.142, *p* = 0.032), AP1 (HARM effect: F[1,57] = 49.307, *p* = 0.001; ZOL effect: F[3,57] = 19.014, *p* = 0.001), and Distance (HARM effect: F[1,57] = 5.421, *p* = 0.023; ZOL effect: F[3,57] = 4.481, *p* = 0.007). LSD post-hoc test: ** *p* ≤ 0.01, *** *p* ≤ 0.001.

**Figure 5 biomolecules-13-00197-f005:**
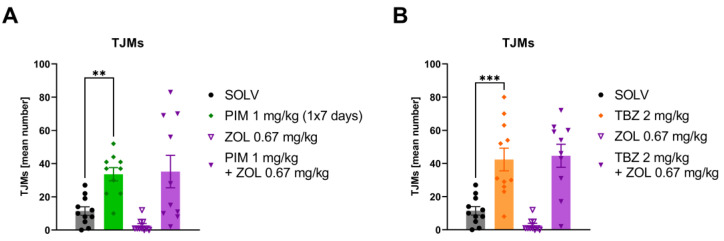
Zolpidem (0.67 mg/kg) had no effect on pimozide-induced (**A**) or TBZ-induced (**B**) tremulous jaw movements (TJMs). The data are shown as the mean (±SEM) number of tremulous jaw movements (TJMs) per 10 min observation period. The number of animals: SOLV, *n* = 11; PIM 1 mg/kg, *n* = 10; TBZ 2 mg/kg, *n* = 11; ZOL 0.67 mg/kg, *n* = 10; PIM 1 mg/kg +ZOL 0.67 mg/kg, *n* = 10; TBZ 2 mg/kg + ZOL 0.67 mg/kg, *n* = 10. Statistics: Factorial ANOVA for PIM (PIM effect: F[1,37] = 26.019, *p* = 0.001; ZOL effect: F[1,37] = 0.446, *p* = 0.508) and TBZ (TBZ effect: F[1,38] = 51.114, *p* = 0.001; ZOL effect: F[1,38] = 0.410, *p* = 0.526) + LSD post-hoc test (** *p* ≤ 0.01, *** *p* ≤ 0.001).

## Data Availability

Research data are available upon request to the first or corresponding author at mroz@if-pan.krakow.pl or kuter@if-pan.krakow.pl.

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
