# Peer review of "GABA-A Alpha 2/3 but Not Alpha 1 Receptor Subunit Ligand Inhibits Harmaline and Pimozide-Induced Tremor in Rats"

_biomolecules, 2023, doi:10.3390/biom13020197_

Round 1

Reviewer 1 Report

The study was a search for GABA-A subunit-specific tremorolytics using different tremor-generating mechanisms. The novel GABA-A α2/3 selective MP-III-024 significantly reduced both the harmaline-induced ET tremor and pimozide-induced TJMs. While zolpidem decreased locomotor activity of the rats, MP-III-024 produced small increases. These results provide important new clues into tremor suppression mechanisms initiated by the enhancement of GABA-driven inhibition in pathways controlled by α2/3 but not α1 containing GABA-A receptors. Potentially GABA-A α2/3 could become novel drug-target tested towards PD and ET tremor therapy with additional potential towards movement facilitation in PD.

 1. (Fig.1) The author administered zolpidem and MP-III-024 in that order. What was the basis for the observation period following administration? Is it a combination of the best of both worlds? Is it necessary to calculate the half-lives of the two drugs?

2. 281-283,The authors tested three doses: low, medium, and high. The dose-dependent concentration of the drug, however, was not determined in this study. Why aren't drug concentrations dose-dependent?

3. 311-313, The authors observed the drug's effect 60 minutes after injecting it into experimental animals. Why do they say the drug has the greatest effect after 60 minutes?

4.403-405, The study's findings did not show the study on these 9 patients. When a drug is combined with the GABAA receptor, it can have two effects on neurons: positive allosteric (PAM) modulator or negative allosteric modulator (NAM). The drug in this study should be PAM, whereas the alcohol consumed by these nine people should be NAM. Forward and reverse regulation do not have the same mechanism. What is the author's reasoning for explaining reverse regulation?

5.The findings of this study shed light on the mechanism of tremor generation. The drug MP-III-024 has an effect on ET and PD tremors. Only by observing the movement of experimental animals can the drug be shown to relieve tremor, but the specific mechanism of action of the drug is unknown.

6.This article mainly explained that combining the drug with GABAA alpha 2/3 site has a certain effect on tremor, but how to verify that the target is GABAA alpha 2/3 if GABAA alpha 1 is excluded? The findings are inconclusive.

Author Response

Thank you for the helpful comments on our manuscript.

The study was a search for GABA-A subunit-specific tremorolytics using different tremor-generating mechanisms. The novel GABA-A α2/3 selective MP-III-024 significantly reduced both the harmaline-induced ET tremor and pimozide-induced TJMs. While zolpidem decreased locomotor activity of the rats, MP-III-024 produced small increases. These results provide important new clues into tremor suppression mechanisms initiated by the enhancement of GABA-driven inhibition in pathways controlled by α2/3 but not α1 containing GABA-A receptors. Potentially GABA-A α2/3 could become novel drug-target tested towards PD and ET tremor therapy with additional potential towards movement facilitation in PD.

  1. (Fig.1) The author administered zolpidem and MP-III-024 in that order. What was the basis for the observation period following administration? Is it a combination of the best of both worlds? Is it necessary to calculate the half-lives of the two drugs?

Author response: Based on both Reviewer’s suggestions we rearranged figure 1 to make it more clear.  Each treatment was depicted separately. Zolpidem and MP-III-024 were administered in separate experiments. The pretreatment times and behavioral observation periods were chosen based upon previous in vivo studies with these compounds - Fischer et al., 2017 for MP-III-024, Trenque T et al. 1994 (doi: 10.1111/j.2042-7158.1994.tb03868.x.), Mierzejewski et al. 2016 for zolpidem and Kosmowska et al. 2020 (doi: 10.1007/s11064-020-03010-5)doi: 10.1016/j.neuroscience.2019.12.045), Kosmowska et al 2017 (doi: 10.1111/cns.12692), Kosmowska et al. 2016 (doi: 10.1111/cns.12467), Ossowksa et al. 2015 (doi: 10.1016/j.pharep.2014.11.008) for harmaline, Salamone et al. 1998 (doi:10.1016/S0301-0082(98)00053-7), Collins-Praino, et al.  2011 (doi:10.3389/fnsys.2011.00049), Podurgiel et al.(doi:10.1016/j.bbr.2015.11.008), Kosmowska et al 2022 (doi: 10.1016/j.bbr.2021.113585) for pimozide and tetrabenazine as well as previous unpublished data – see supplementary results. Compound half-life, distribution rate due to the administration way, as well as the experimental observations of occurrence of first symptoms, the fact that the target is an ionotropic, fast acting receptor and length of effect (as for harmaline) were taken into consideration while choosing the times of administration and observation.

  1. 281-283,The authors tested three doses: low, medium, and high. The dose-dependent concentration of the drug, however, was not determined in this study. Why aren't drug concentrations dose-dependent?

Author response: Doses were chosen based upon prior in vivo studies with these compounds (see Methods). Zolpidem was tested in three doses, while MP-III-024 was tested in two doses.

Based on the pharmacokinetic studies previously reported for zolpidem (Trenque T et al. 1994 doi: 10.1111/j.2042-7158.1994.tb03868.x.) and our previous studies reporting efficacy of low doses of zolpidem in animal studies (0.3 and 1mg/kg, Mierzejewski et al. 2016), we selected: 1.01 mg/kg; 0.67 and 0.34 mg/kg for the present study. We reasoned that if zolpidem previously showed antipsychotic activity at 0.3 and 1mg/kg – the concentration of the drug in plasma should be sufficient to display potential antitremor efficacy. Since zolpidem can cause sleepiness we aimed at low, but still effective doses. In order to definitively exclude the anti-tremor effect we tested 3 doses to determine the dose impacting locomotor activity. Considering the above, there was no need to investigate the concentrations of a drug following administration of all tested doses.

As for MP-III-024 - this was exploratory experiment to check the working hypothesis on GABAAalpha2/3 subunit receptor as an anti-tremor target, not to evaluate its pharmacokinetics. This was done for previous nociceptive studies in mice. For tremor, this work will be performed in subsequent studies as well as testing the time-course of the effects. Moreover, in order to comply with 3Rs rule and to minimalize the number of animals in the study (recommended by European Union Directive of 22 September 2010 (2010/63/EU), we did not evaluate the drug-dependent concentration of the drug.

We have also added new Discussion to the issue of lack of dose-dependency. See in line 438 - .445 “MP-III-024 produced significant decreases in tremor but these effects were not always dose-dependent….”

  1. 311-313, The authors observed the drug's effect 60 minutes after injecting it into experimental animals. Why do they say the drug has the greatest effect after 60 minutes?

Author response: Good point. As described in Methods and Fig 1. in the experiment with zolpidem and harmaline, we analyzed tremor and locomotor activity for 30 minutes while in the experiment with MP-III-024 and harmaline we performed this observation longer (up to 60 min) in order to have a better first look at the time-course of the effect. It gave us opportunity to compare the effects of the two time-points 0-30 min vs 30-60min. This is important in the comparison to the harmaline which shows dual effects on locomotor activity – first it decreases it while later increases it. Usually animals explore during the first 15-30 min and then settle and just sit (as in controls here). Therefore, it seems that MP-III-024 has a slightly normalizing effect on harmaline in this aspect.

  1. 403-405, The study's findings did not show the study on these 9 patients. When a drug is combined with the GABAA receptor, it can have two effects on neurons: positive allosteric (PAM) modulator or negative allosteric modulator (NAM). The drug in this study should be PAM, whereas the alcohol consumed by these nine people should be NAM. Forward and reverse regulation do not have the same mechanism. What is the author's reasoning for explaining reverse regulation?

Author response: We are sorry for the confusion.  We are not trying to discuss reverse regulation here but to show that GABAA alpha 2/3 could be a better antitremor GABAA-mediated therapy than alcohol intoxication. TPA023 is alpha2/3/5 selective. Ethanol was discussed here in the sense of a positive control (reducing tremor). EtOH has many biological effects beyond GABA. Yet many studies show that EtOH potentiates GABAA. See elsewhere: “Ethanol, at concentrations ranging from 1 to 50 mM, potentiated GABAA responses of acutely dissociated neurons from rat neocortical slices and primary neuronal cultures from chick, mouse and rat brain (Reynolds et al., 1992). GABAA-activated chloride currents were also shown to be potentiated by ethanol in cultured mouse hippocampal and cortical neurons (Aguayo, 1990; Reynolds and Prasad, 1991). At the single channel level, ethanol enhanced the frequency of GABA-mediated channel opening events, mean open time, open time percentage, frequency of opening bursts, and mean burst duration (Tatebayashi et al., 1998). “

We have deleted the part discussion on EtOH to avoid confusion.  Thank you.

  1. The findings of this study shed light on the mechanism of tremor generation. The drug MP-III-024 has an effect on ET and PD tremors. Only by observing the movement of experimental animals can the drug be shown to relieve tremor, but the specific mechanism of action of the drug is unknown.

Author response: Yes, this is a preliminary report which could be the starting point to search for network-, brain stricture-, cell-type, molecular pathway-specific mechanisms.

  1. This article mainly explained that combining the drug with GABAA alpha 2/3 site has a certain effect on tremor, but how to verify that the target is GABAA alpha 2/3 if GABAA alpha 1 is excluded? The findings are inconclusive. 

Author response: This is a key question, thank you for raising this point. If the compound acting selectively on alpha 2/3 GABA-A receptors -displays anti-tremor efficacy and the selective alpha 1 GABA-A PAM- does not - then we can conclude that alpha 2/3 potentiation is sufficient and that alpha 1 potentiation is not.  Definitive linkage of effect to mechanism will involve future planned studies: investigation of other GABA-A alpha2/3 selective agonists, blockade of alpha 1 effects of non-selective GABA PAMS (e.g., diazepam) or to use GABA-A alpha2/3 genetically -deprived animals. The present manuscript represents a preliminary report identifying the antitremor efficacy of GABA-A alpha 2/3 PAM as a new potential target for further research. We plan to conduct a follow-up studies for this interesting compound, however, these studies are currently beyond the scope of the present manuscript and will addressed in the future.

We have now added some discussion points to help answer this question. See line 420. “That α2/3 but not α1 subunit-containing GABA-A receptors are sufficient for anti-tremor efficacy is based on…”

Reviewer 2 Report

In this work, the authors did a potentially interesting investigation about the role of MP-III-024, rather than zolpidem, in the treatment of transient tremors. I greatly appreciate the effort of the authors to perform such a complex pharmacological and behavioral analysis. However, due to the complex nature of the experimental design performed, the way the manuscript was constructed is very hard to understand the different paradigms and therefore, the conclusion of the experiments. Please find the concerns raised below:

1)    The authors provided a detailed background introduction into tremor, especially regarding essential tremor and PD tremor; however, the harmaline-induced tremor model is mostly a transient and acute model of tremor, especially in the experimental design used, and therefore most likely does not induce the relevant neuroplasticity of the syndromes introduced. In a similar manner, the tremor induced using pimozide and tetrabenazine also have particular effects that lead to the development of these symptoms and should be addressed in the introduction. Authors should consider that Parkinson’s disease is a much more complex disorder than “only” dopaminergic denervation caused by these drugs.

2) The experimental design and results are presented in an odd manner that makes it difficult to understand the aim and purpose of each experiment. Even though figure 1 is very well constructed, the order of the results makes it challenging to understand the overall paradigm investigated and take conclusions. Authors need to align the experimental design presentation with the way results are presented. For example, the Scheme A of Figure 1, makes the author believe that the effect of zolpidem and MP-III-024 will be compared after the harmaline-induced tremor. However, the authors chose to show the effect of ZOL in all the models first, making if somewhat confusing to the reader.

3)    Because this is a very interesting article regarding behavioral aspects, authors should present the individual values for each animal (data point).

4)    Considering that the authors suggest that the use of pimozide and tetrabenazine would mimic the parkinsonism tremor, do the authors evaluate motor deficits (open field test, rotarod, or similar) in these animals? Please consider that these results are very interesting regarding the effect of MP-III-024 in all the models of transient tremor. As discussed by the authors, treatments that modulate GABAergic systems often induce important side effects. It would be interesting, going forward, that the authors investigate the safety of MP-III-024 in these side effects, not only in the harmaline-induced tremor but also in the dopaminergic-depletion models.

5)    I strongly suggest that the authors describe each graph in the figure with a different letter (for example, figure 2A to 2E) to better understand each result presented.

Author Response

Thank you for the helpful comments on our manuscript.

In this work, the authors did a potentially interesting investigation about the role of MP-III-024, rather than zolpidem, in the treatment of transient tremors. I greatly appreciate the effort of the authors to perform such a complex pharmacological and behavioral analysis. However, due to the complex nature of the experimental design performed, the way the manuscript was constructed is very hard to understand the different paradigms and therefore, the conclusion of the experiments. Please find the concerns raised below:

  • The authors provided a detailed background introduction into tremor, especially regarding essential tremor and PD tremor; however, the harmaline-induced tremor model is mostly a transient and acute model of tremor, especially in the experimental design used, and therefore most likely does not induce the relevant neuroplasticity of the syndromes introduced. In a similar manner, the tremor induced using pimozide and tetrabenazine also have particular effects that lead to the development of these symptoms and should be addressed in the introduction. Authors should consider that Parkinson’s disease is a much more complex disorder than “only” dopaminergic denervation caused by these drugs.

  • The experimental design and results are presented in an odd manner that makes it difficult to understand the aim and purpose of each experiment. Even though figure 1 is very well constructed, the order of the results makes it challenging to understand the overall paradigm investigated and take conclusions. Authors need to align the experimental design presentation with the way results are presented. For example, the Scheme A of Figure 1, makes the author believe that the effect of zolpidem and MP-III-024 will be compared after the harmaline-induced tremor. However, the authors chose to show the effect of ZOL in all the models first, making if somewhat confusing to the reader.

Author response: Thank you for pointing this out. Figure 1 was reorganized now showing each experiment separately. The sequence of methods, results and discussion was aligned to have more transparent form.

  • Because this is a very interesting article regarding behavioral aspects, authors should present the individual values for each animal (data point).

Author response: All results figures were adapted and show individual animal values as points on the graphs.

  • Considering that the authors suggest that the use of pimozide and tetrabenazine would mimic the parkinsonism tremor, do the authors evaluate motor deficits (open field test, rotarod, or similar) in these animals? Please consider that these results are very interesting regarding the effect of MP-III-024 in all the models of transient tremor. As discussed by the authors, treatments that modulate GABAergic systems often induce important side effects. It would be interesting, going forward, that the authors investigate the safety of MP-III-024 in these side effects, not only in the harmaline-induced tremor but also in the dopaminergic-depletion models.

Author response: The pimozide and tetrabenazine models were used as based on previous publications of Salamone’s group and our own unpublished observations by prof. J. Wardas and dr B. Kosmowska. We show in the added supplementary materials that both pimozide (7 daily doses, 1 mg/kg ip) as well as tetrabenazine (1 dose, 2 mg/kg ip) significantly reduce (by approx. 50%) walking distance of Wistar rats as measured in the same FPA cages used to analyze tremor. This effect lasted until 35 and 40 minutes respectively, until animals settled and stopped exploring. Both substances also induced catalepsy. See details in the attached supplement. Those were preliminary experiments to establish tremor models. Drugs were not tested in such way after pimozide or tetrabenazine because TJM are not picked up by FPA, in contrast to the whole-body tremors induced by harmaline.

We have reported here on locomotor activity as a potential measure of side-effects caused by zolpidem. MP-III-024, both in our study and in the work of Fischer et al. (2017) discussed here, was shown to produce little in the way of such side effects.  Indeed, it has been noted that the imidazodiazepine series of compounds has nicely shown large separations in efficacy vs. side effects (see e.g., Witkin et al., 2022 for summary). The aspect of side effects will be in focus of future research. Thank you for this suggestion.

  • I strongly suggest that the authors describe each graph in the figure with a different letter (for example, figure 2A to 2E) to better understand each result presented.

Author response: All graphs and figures were named as suggested.

Round 2

Reviewer 2 Report

The authors sufficiently answered all concerns raised.